# *Robot See Robot Do:* Imitating Articulated Object Manipulation with Monocular 4D Reconstruction

**Justin Kerr**[*]    **Chung Min Kim**[*]    **Mingxuan Wu**    **Brent Yi**
**Qianqian Wang**    **Ken Goldberg**    **Angjoo Kanazawa**
[*]Equal Contribution
UC Berkeley
https://robot-see-robot-do.github.io

**Abstract:** Humans can learn to manipulate new objects by simply watching others; providing robots with the ability to learn from such demonstrations would enable a natural interface specifying new behaviors. This work develops *Robot See Robot Do* (RSRD), a method for imitating articulated object manipulation from a single monocular RGB human demonstration given a single static multi-view object scan. We first propose 4D Differentiable Part Models (4D-DPM), a method for recovering 3D part motion from a monocular video with differentiable rendering. This analysis-by-synthesis approach uses part-centric feature fields in an iterative optimization which enables the use of geometric regularizers to recover 3D motions from only a single video. Given this 4D reconstruction, the robot replicates object trajectories by planning bimanual arm motions that induce the demonstrated object part motion. By representing demonstrations as part-centric trajectories, RSRD focuses on replicating the demonstration's *intended* behavior while considering the robot's own morphological limits, rather than attempting to reproduce the hand's motion. We evaluate 4D-DPM's 3D tracking accuracy on ground truth annotated 3D part trajectories and RSRD's physical execution performance on 9 objects across 10 trials each on a bimanual YuMi robot. Each phase of RSRD achieves an average of 87% success rate, for a total end-to-end success rate of 60% across 90 trials. Notably, this is accomplished using only feature fields distilled from large pretrained vision models — without any task-specific training, fine-tuning, dataset collection, or annotation. Project page: https://robot-see-robot-do.github.io

**Keywords:** Visual Imitation, Articulated Objects, Feature Fields

## 1 Introduction

Consider teaching a robot to manipulate an articulated object in your house such as a pair of scissors or sunglasses. The most natural way to do this is simply to pick up the object, show it to the robot, and then demonstrate how to use it with your own hands. This is how children learn — from observing adults — despite the cross-morphology gap between the large hands of adults and the small hands of a child. A key insight that enables visual imitation across a morphology gap is not to focus on the exact motion of the *manipulator* (i.e., hand), but observe the consequence of the action at the *object* level. If the 3D motion of an object and its parts can be perceived, one could plan to manipulate the object such that the perceived 3D motion to be replicated.

This paper proposes Robot See Robot Do, an object-centric method for manipulating objects with moveable parts from a single human demonstration given 1) a static multi-view object scan and 2) a *monocular* human interaction video. These inputs are easily captured with any smartphone. The **See** phase builds a model of the object, groups it into movable parts, and recovers their 3D motion trajectories. During robot deployment in the **Do** phase, the robot is presented with the same object in an unknown pose in the workspace. The robot registers the recovered 3D object trajectory from the

8th Conference on Robot Learning (CoRL 2024), Munich, Germany.

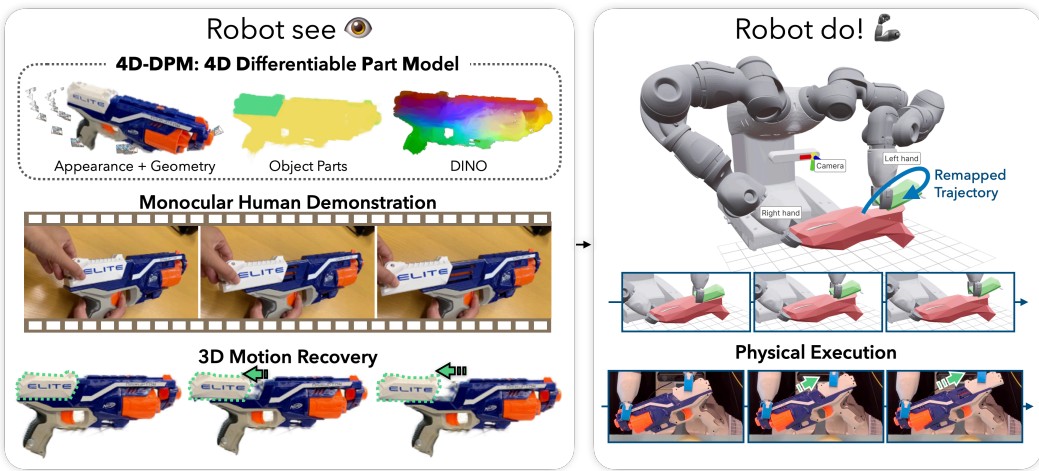

Figure 1: **Robot See Robot Do.** To visually imitate articulated object motion RSRD first reconstructs a part-aware feature field. Given an input demonstration video, we then track the object part motion using the feature field. Next, the robot recognizes the object in its workspace and plans a bimanual trajectory to achieve the demonstrated object motion.

demonstration to the pose of the object in the world, then plans a bimanual end-effector trajectory to induce the same 3D motion on the object as perceived in the demonstration video. Because the demonstrations are recovered in an object-centric manner, the same demonstration can be reused for different robots, grippers, and re-orientations of the object.

Recovering the 3D movement of an object and its parts from a monocular video is challenging due to the under-constrained nature leading to degenerate solutions. In this paper, we propose 4D-Differentiable Part Models (4D-DPM), a method which uses a decomposed 3D feature field to recover part and object motion from monocular videos. 4D-DPM leverages an *analysis-by-synthesis* paradigm, where a model of 3D part motion is iteratively compared to visual observations and fit through optimization. 4D-DPM first processes the multi-view static video of an object with GARField [1] to construct a 3D Gaussian Splat [2] segmented into parts. Then, it embeds DINO [3] feature fields into each object part, which enables tracking the object motion in a monocular video by comparing it to video-computed DINO features through differentiable rendering. By leveraging the 0-shot performance of visual representations in large pretrained models, 4D-DPM enables tracking a wide variety of objects without any fine-tuning or task-specific dataset collection. Additionally, 4D-DPM can naturally incorporate any prior one can represent with a differentiable loss function; for example temporal smoothness and as-rigid-as-possible prior.

During deployment, RSRD generates a set of candidate grasps for moving each desired subpart, then finds a collision-free bimanual motion that rigidly tracks the motion of grasped objects throughout the trajectory. To determine which parts to grasp, we recognize hand-part contacts in the demonstration video and softly bias robot grasps towards these parts. Notably, RSRD does not attempt to copy the motion of human hands, allowing it to find robot motions which achieve the same object trajectory with different embodiment.

We evaluate RSRD on a variety of 9 articulated objects, ranging from tools to plushies, assessing its flexibility to function on a diverse range of objects. Notably, several demonstrations are accomplished with bimanual manipulation, fully lifting the object off the workspace. For tracking, RSRD achieves an average distance error of 7.5 mm compared to ground truth part poses; ablations highlight the importance of both as-rigid-as-possible regularization and DINO for successful tracking. For real-world robot experiments, we employ a bimanual YuMi robot to measure success rates across four distinct phases of the RSRD robot execution pipeline, with each object placed in 10 different orientations within the robot's workspace. The results demonstrate a success rate of 94% for initial pose registration and 87% for trajectory planning. Initial grasps and motion execution recorded success rates of 83% and 85% respectively; end-to-end, this means that RSRD achieves successful

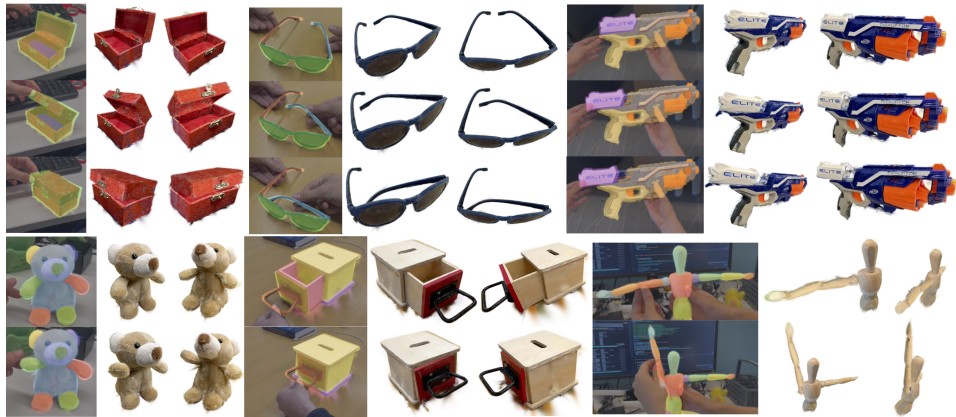

Figure 2: **4D Reconstruction of Articulated Objects**. Keyframes from the motion trajectories overlaid over monocular RGB demonstrations with parts colorized, and along with two viewpoints. See more qualitative results on the website.

imitation for 60% of initial object positions. Importantly, these results are achieved solely through feature fields derived from pretrained vision models, without relying on any task-specific training, fine-tuning, data collection, or annotation.

## 2   Related Work

**Recovering 3D Motion for Objects with Moving Parts.**   Reconstructing the 3D motion trajectory of articulated object from a single video is extremely challenging as it involves detecting and reconstructing individual parts and recovering their poses across space and time. A substantial body of work bypasses the reconstruction problem and utilizes point clouds to perceive articulated parts, with inputs ranging from sequences of articulated motions of objects [4, 5, 6, 7] to single point clouds [8, 9, 10, 11, 12, 13]. More similar to our approach are works that take in visual observations as input for joint reconstruction and part segmentation. Given the challenging nature of modeling moving objects, most work require either RGB-D videos or multi-view observations at multiple states [14, 15, 16, 17] as input. There exist monocular based articulated object tracking methods, but they typically require known kinematic chains [18, 19] or category-specific priors [20, 21]. In contrast, after seeing the object once, RSRD functions from *purely monocular* interaction input video. In addition, unlike previous work that relies on training in small-scale datasets with part-level annotations [22], we distill the segmentation of the parts from SAM [23] into 3D using GARField [1], which generalizes well to objects in the wild.

**Learning from one demonstration.**   Human videos are valuable resources for learning object interaction behaviors. Extensive research [24, 25, 26, 27] has leveraged human video data to learn robot manipulation, but techniques largely still require additional robot teleoperation data for target tasks or paired human-robot data to bridge the morphology gap. Also related to our method are works that learn manipulation policies from a single demonstration [28, 29, 30, 31, 32, 33], many of which learn from humans [34, 35, 36]. While these methods enable a robot to perform tasks from one demonstration, they require extensive in-domain data and well-curated meta-training tasks for training, which limits the generalization of the learned policy. In contrast, our method enables manipulation from a single human video using an object-centric formulation, without requiring extensive training with in-domain data. Our setting only requires an additional multiview capture to obtain the 3D scan of the manipulated object, which can be achieved with a smartphone camera.

**Object-centric representations for robot manipulation.**   Object-centric representations have enabled a broad range of capabilities for robot manipulation. Many existing works have focused on how representations for objects can be learned from specific classes of data, for example by using 3D structure as a signal for contrastive learning [37], intermediate object properties as a bot-

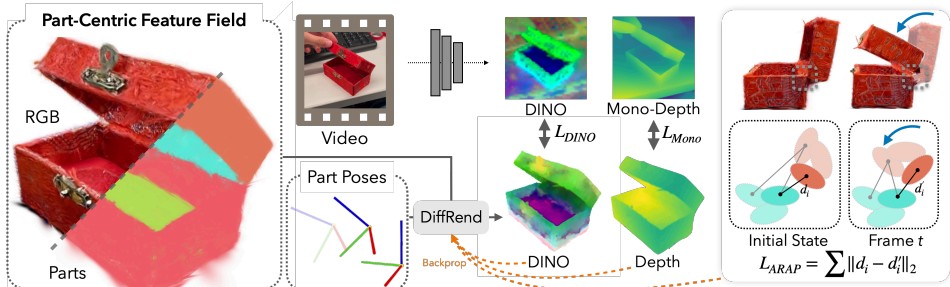

Figure 3: **4D Differentiable Part Models (4D-DPM)**. *Left*: DINO features and depth are rendered from per-timestep optimizable part pose parameters, and compared with extracted DINO features and monocular depth from the input frame. *Right*: an ARAP loss penalizes gaussians from deviating too far from their initial configuration with respect to neighbors. Together these losses flow backwards into the part poses and are optimized with gradient descent to recover 3D part motion.

tleneck for image prediction [38], or canonical object views as conditioning [39]. Others have shown how object-centric approaches can be used to improve generalization in robot manipulation via imitation [40, 41, 32, 33, 42, 43] or reinforcement [44, 45] learning, as well as to augment robots with semantic [46], relational [47, 48, 49, 50], uncertainty-based [51, 52, 53], symbolic [54, 55, 56, 57, 58, 59], and part-aware [60, 61] reasoning. In RSRD, we show how relative motion tracked via *part-centric* representations can be used for single-shot imitation. Importantly, these representations do not require fixed object categories or task-specific data. Instead, we rely on and reap the open-world benefits of large pretrained models.

**Feature fields for robotics.** 3D neural fields have recently been explored in robotics, beginning with exploring leveraging Neural Radiance Fields [62] (NeRFs) as as high-quality visual reconstruction for grasping [63, 64] and navigation [65], and more recently by leveraging its ability to embed higher dimensional features for language-guided manipulation [66, 67]. A core limitation of neural fields is their slow training speed, an issue which is ameliorated by 3D Gaussian Splatting (3DGS) [2], a technique for representing radiance fields as a collection of oriented 3D gaussians which can be differentiably rasterized quickly on modern GPUs. Concurrent works transfer high-dimensional feature fields to 3DGS for rapid training and rendering, as well as language-guided robot grasping [68, 69]. One remaining core limitation is that these representations are static, and must be re-scanned after moving the environment. Wang et al. [70] showed promising results on tracking DINO embedded object keypoints in 3D from multi-view cameras. In this work, we develop a method for recovering 3D motion of 3DGS feature fields from a monocular video.

## 3  Problem and Assumptions

Given a bimanual robot with parallel-jaw grippers, a multi-view object scan of an object with two or more movable parts, and a monocular human demonstration, the goal is to manipulate the object through the same configuration change starting from an unknown location in the robot's workspace. We focus on articulated objects with one or more rotary or prismatic joints, and assume that internal part deformation is negligible. We also assume the object scan is taken in the same starting configuration as the demonstration, and that the input video has a static viewpoint with clear visibility of the subpart being manipulated.

## 4  Method

RSRD first builds a 4D Differentiable Part Model of the object segmented into parts embedded with feature descriptors (Sec 4.1), and uses these dense part descriptors to recover 3D motion from a monocular video (Sec 4.2). Next, during deployment the robot recognizes the pose of the object in its workspace and plans actions which emulate the human's to bring it through the same range of motion (Sec. 4.3).

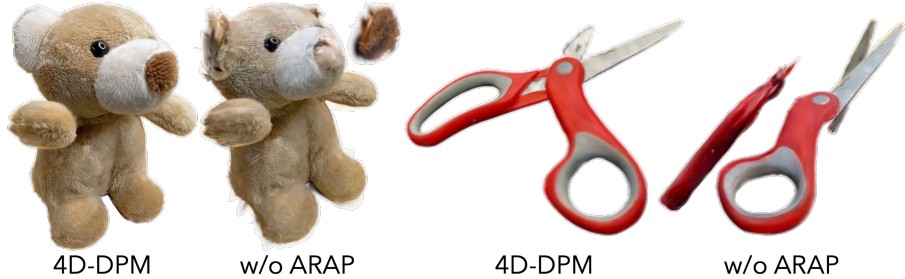

| 4D-DPM | w/o ARAP | 4D-DPM | w/o ARAP |

Figure 4: **ARAP Ablation.** ARAP is a simple but effective prior for improving 3D motion recovery by preventing small or under-observed parts from drifting.

## 4.1 Constructing 4D Differentiable Part Models

Given a static multi-view capture of the object of interest, we first construct a 3D model for the object using Gaussian Spatting (3DGS). We leverage 3DGS because it has been shown to be orders of magnitude faster in reconstructing and rendering both visual appearance and high-dimensional feature fields [2, 69], while in addition providing an explicit representation that can be easily segmented into objects and subparts. In parallel, we train a GARField [1] from the same capture. GARField can cluster the 3D Gaussians into discrete groups of varying granularities, controlled by a scale parameter. This allows manual segmentation of the 3D object from the background by clicking it in the scene, and manual decomposition of the object into parts by selecting a scale parameter at which to break apart the object where all relevant parts are separated. For more implementation details please see the Appendix.

## 4.2 Monocular 3D Part Motion Recovery

Recovering 3D motion of object parts from a single RGB video is a highly underconstrained and challenging problem. To tackle it, we propose an analysis-by-synthesis approach (Fig. 3). Instead of feed-forward inference we optimize, or *synthesize*, a model of the object parts over time, to understand (i.e *analyze)* their motion. 4D part pose over time is represented as a trajectory of SE(3) poses per time-step, where each consecutive timestep is initialized from the previous. This approach leverages the differentiable rendering of feature fields to backpropagate pixel errors into 3D pose deltas. This is appealing over feedforward tracking methods as it can be integrated with sophisticated regularizations like geometric rigidity or temporal smoothing through optimization, as we leverage in this work. In addition, 4D-DPM's use of large pretrained vision models lends robustness to a large diversity of objects zero-shot.

**Part Motion Optimization from Video**   To obtain pose updates for an object's parts given a new frame, we render the 4D-DPM at a virtual camera with the same intrinsics as the input video to obtain rendered outputs including RGB images, depth maps, and DINO feature maps. In parallel, we extract DINO features from the input video frame, and compare them directly to the rendered features with an MSE loss. Because rendering is fully differentiable with respect to the part poses, by backpropagating through the entire rendering process the poses of individual object parts can be optimized with gradient descent (Fig 3). We experimentally validate that DINO features offer a much more robust optimization target than photometric loss, and so all experiments use DINO instead of photometric (Tab 2).

**3D Regularization Priors**   Reliably tracking pose from pure monocular RGB is a significant challenge because of depth ambiguity. 4D-DPM uses the fact that all object parts can be jointly optimized, allowing for regularizing the optimization with external 3D priors using auxiliary losses. We use two 3D priors: a regularization from a mono-depth prediction network and a local as-rigid-as-possible (ARAP) penalty. These losses are added to the primary DINO loss described above, please see the Appendix for hyperparameters. The first regularization $L_{mono}$ imposes a soft constraint towards outputs from Depth Anything [71]. To account for the fact that mono-depth output

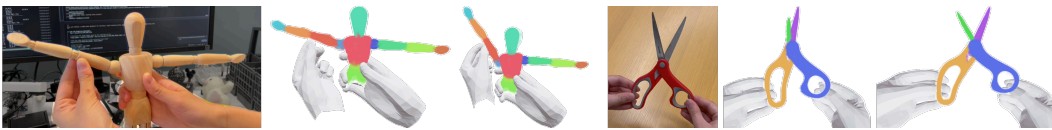

Figure 5: **Hand Alignment**: RSRD uses HaMeR [74] to detect and align human hand poses to the demonstrations. Detections are used to rank part pairs for grasping (Sec 4.3).

is non-metric, we use the ranking-based loss proposed in Sparse-NeRF [72]. Specifically, we sample pairs of points within the rendered object mask and enforce their relative depth orders between our rendered depth and the mono-depth to match. The object mask is eroded by 5 pixels to reduce sensitivity to misalignment.

The second loss is an adaptation of as-rigid-as-possible (ARAP) [73] loss, which is only applied to boundary gaussians between parts. We compute $L_{\text{ARAP}}$ by finding boundary gaussians between each pair of parts, defined by a radius threshold of 2.5 mm on their centers, and storing the initial distance between neighboring pairs $d_{\text{init}}$. During optimization we impose a loss penalizing gaussians from drifting away from that initial distance $\sum_{ij} \rho(d_{\text{init}}^{ij} - d_{\text{current}}^{ij})$. $L_{\text{ARAP}}$ does not penalize neighboring gaussians from rotating with respect to one another, allowing hinging movement easily.

**Initialization**    During robot execution and for the first frame of each demonstration video, 4D-DPM must estimate the object's SE(3) pose either for manipulating the object, or initializing the 3D motion estimation for subsequent frames. To initialize the object pose for a single frame, RSRD first approximately locates the object in the 2D image. To do this, we compare the object's 3D DINO features to the frame's 2D DINO features, finding mutual nearest neighbors between 3D gaussian and pixel features. The pixel centroid of these matches creates a ray in 3D space, along which we place the centroid of the object in 3D at a fixed distance from the camera. 4D-DPM then executes 8 seeds of object pose optimization for 200 iterations, rotating about the object's gravity axis, and select the pose with lowest loss. During robot execution, the exact same procedure is used with stereo depth instead of monocular depth.

## 4.3   Object Motion and Grasp Planning

Once the poses of the object parts are registered in the world frame of the robot, we plan feasible robot trajectories to impart the desired motion onto the object. This takes place in three stages: 1) part selection, to decide which parts should be moved; 2) grasp planning, to decide which parts are kinematically reachable by the robot; and 3) trajectory planning, to decide which parts can be manipulated through the full part trajectory performed in the demonstration.

**Hand-Guided Part Selection**    We first create a list of candidate parts for the robot to interact with, by estimating which ones the human hand interacts with. Note the naive method of choosing the maximally-traveled part will fail if object parts are coupled. For example, for *"opening the scissors"*, where all parts move the same distance, the motion cannot actuate the scissors if the chosen parts are the scissor blade and handle rigidly connected to each other. Biasing with the hand helps avoid these degenerate pairs. We emphasize these detections are only used to bias part selection, but not grasps, since hands can perform grasps impossible with a parallel-jaw gripper.

To detect the human hands in 3D, we use HaMeR [74] to detect hand pose meshes. We calculate the hand's metric size and pose by matching it to the estimated metric depth of the hand, which we calculate by scaling and shifting the image monodepth by the rendered object gaussian depth. Then, we register part-hand interactions by computing the part-hand assignments which globally minimizes thumb and index finger distance to the given parts across the trajectory. Finally, a ranked list of actuated object parts is calculated based on the part distance metric.

**Part-Centric Grasp Planning**    Before we ask a robot to reliably execute object motions, we must first account for the limitations of a robot parallel-jaw gripper. Mainly, the human hand can flexibly

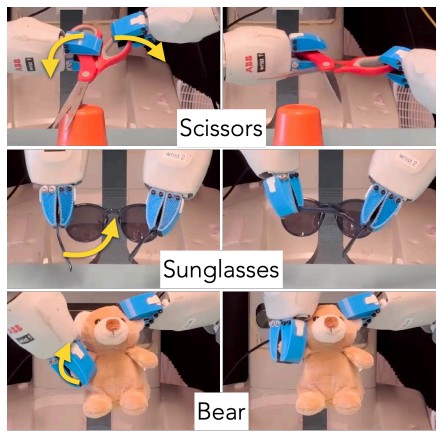

Figure 6: **Example Robot Executions**. Arrows indicate direction of motion.

| Objects | Init. | Traj. | Grasp | Exec. |
|---|---|---|---|---|
| Red Box | 10/10 | 9/10 | 8/9 | 4/8 |
| Nerf Gun | 10/10 | 10/10 | 8/10 | 7/8 |
| Scissors | 9/10 | 7/9 | 5/7 | 5/5 |
| Sunglasses | 8/10 | 8/8 | 7/8 | 7/7 |
| Bear | 10/10 | 7/10 | 6/7 | 6/6 |
| Stapler | 9/10 | 9/9 | 7/9 | 7/7 |
| Light | 10/10 | 10/10 | 8/10 | 6/8 |
| Wirecutter | 10/10 | 10/10 | 7/10 | 6/7 |
| USB Plug | 8/10 | 7/8 | 7/7 | 6/7 |
| Total | 84/90 | 77/84 | 63/77 | 54/63 |

Table 1: **Physical Trials**. We report success of individual stages of the RSRD pipeline: object pose initialization, trajectory planning, grasp execution, and motion execution.

slide around an object part with controlled contact, or use prehensile motion or wide grabs – which might not be possible with a robot. Thurs, it is important that the robot stays rigidly attached to the object part throughout the trajectory, and every part must have viable grasps, should that part need to be manipulated. We analytically generate part-centric grasps by sampling antipodal grasps on the meshified parts, from which we calculate the target EE pose. See appendix for more details.

**Robot Trajectory Planning**  Given the candidate list of parts, part-centric grasps, and part motions, we exhaustively search for collision-free, kinematically feasible robot trajectories. We first create a list of parts $[p_1, p_2, ...]$, or a list of part-part pairs $[(p_1, p_2), ...]$ for bimanual tasks. Then, for each candidate part(s), we generate a list of robot end-effector pose motions, each starting from one of the 480 part-centric grasps. We implement a sparse Levenberg-Marquardt solver that performs trajectory optimization for each part motion. Trajectories that deviate too far from the target end-effector poses are rejected. For the remaining trajectories, we use cuRobo [75] for collision avoidance checks and to plan robot approach motions. We return the first successful trajectory for physical execution. In bimanual experiments, the robot lifts the object with both hands 2cm off the workspace to avoid table collisions.

## 5 Experimental Results

For physical execution, we use an ABB YuMi robot because of its 7-DoF bimanual arms, and equip it with soft 3D-printed parallel-jaw grippers from Elgeneidy et al. [76]. Extra compliance from soft caging grasps is helpful for making robot execution less sensitive to error in object tracking. We use a ZED 2 stereo camera for providing depth estimates for more accurate object registration. To capture data for demonstrations we use the Polycam phone scanner, which provides posed cameras with metric scale. We collect demonstrations for articulated objects (Fig. 2) which consist of a human demonstrating a degree of freedom to actuate with one or both hands clearly visible. In this work we use demonstration videos which clearly show the object-hand interaction with simple backgrounds and leave complicated in-the-wild videos to future work.

### 5.1 Demonstration Execution

To test how well RSRD can transfer human demonstrations to a robot, we select 9 articulated objects for the robot to actuate, listed in Tab. 1 and detailed in the Appendix. We run 10 trials for each object on the robot, varying the *z* orientation of the object 360° about its centroid while remaining centered in the robot workspace. This means in half of the experiments the initial pose of the object is flipped as compared to the demonstration video, necessitating 3D object-centric reasoning to perform the task. We measure the success rate of each stage of the pipeline: pose initialization, grasp planning, physically grasping, and executing the motion. The final state is evaluated qualitatively by

| Method | Red Box | Nerf Gun | Toy Drawer | Sunglasses | Frog | Average |
|--------|---------|----------|------------|------------|------|---------|
| 4D-DPM | **8.16**$_{\pm 0.77}$ | **3.37**$_{\pm 0.64}$ | **5.85**$_{\pm 1.52}$ | **4.58**$_{\pm 0.39}$ | **10.10**$_{\pm 2.02}$ | **6.41**$_{\pm 1.07}$ |
| No Depth | 9.20$_{\pm 3.40}$ | 3.71$_{\pm 0.53}$ | 6.87$_{\pm 1.40}$ | 4.66$_{\pm 0.69}$ | 21.43$_{\pm 3.00}$ | 9.17$_{\pm 1.80}$ |
| No ARAP | 13.05$_{\pm 2.55}$ | 4.06$_{\pm 0.49}$ | 7.74$_{\pm 1.45}$ | 13.45$_{\pm 2.13}$ | 17.27$_{\pm 3.00}$ | 11.11$_{\pm 1.92}$ |
| Photometric | 47.14$_{\pm 4.93}$ | 58.87$_{\pm 7.39}$ | 74.23$_{\pm 6.02}$ | 56.34$_{\pm 2.11}$ | 47.09$_{\pm 7.15}$ | 56.73$_{\pm 5.52}$ |

Table 2: **Object Part Pose Tracking Evaluation**. We report object part pose tracking accuracy measured by average point-distance (ADD). RSRD significantly outperforms photometric tracking, owing to its use of DINO features as a robust optimization target.

an experimenter who watches the robot's full motion, and checks if the robot imparted semantically similar part motion as the demonstration video (e.g., "close the sunglasses").

Full results are reported in Table 1. Overall, RSRD can reliably register objects in the correct pose in 84 of 90 trials, after which it plans feasible robot motions for 77 of 84 registered poses. This highlights RSRD's flexibility to object re-orientation within the workspace, reproducing object motions even though the object is mirrored with respect to the input demo. When physically grasping and executing these motions, RSRD succeeds for 63 of 77 final plans. This performance dropoff comes primarily from the very low grasp error tolerance needed to accomplish tasks, with many cases narrowly missing grasps after grazing the object during approach. For the red box, subtle tracking shift causes the demonstration to lift upwards by around 3cm in the workspace, lifting the box in the air and sometimes dropping it on its side rather than upright.

## 5.2   Motion Recovery Ablations

We evaluate part tracking performance on 5 objects by capturing their demonstrations with a stereo camera, then manually annotating the ground truth part pose at select keyframes to match the stereo depth obtained from RAFT-Stereo [77]. By aligning gaussians with the depth obtained from the Zed camera, we ensure the correspondence between the annotated poses and actual depth information. Following Wen et al. [78], we report average point-wise distance (ADD) and compare against 3 ablations of our method: 4D-DPM without ARAP, 4D-DPM without depth regularization, and Photometric only tracking. Results are reported in Table 2. Motion recovery with DINO feature fields strongly outperforms photometric tracking, which diverges severely in most cases because of an inability to distinguish between foreground and background.

## 6   Discussion

**Limitations and Future Work**   The main limitation of RSRD is the assumption that object start configurations match their demonstration, and thus is sensitive to small amounts of difference in initial configurations. Future work will study how to adapt to these cases. In addition, the existence of manual segmentation phase should ideally be automated to increase scalability of the method, perhaps based on the perceived motion in the demonstration. Because the method uses monocular RGB input only, it is also sensitive to the quality of the object scan and the viewing angle of the demonstration video, sometimes struggling in cases where the background in the demonstration is too complicated, overpowering the DINO features on the object. Finally, the tracker struggles with highly symmetric or featureless objects where DINO doesn't provide enough motion cues, or objects with small parts. Robot execution in RSRD currently assumes rigid parallel-jaw grasps, an assumption which would be interesting to lift in future work for planning non-prehensile motions.

**Conclusion**   This paper presents Robot See Robot Do, a method for teaching articulated object motions with a single monocular human demonstration, and replicating them on a bimanual robot. It takes advantages of neural feature fields for building a part-aware model of the object which can be tracked in an input monocular video without the need for labeled part datasets. Because of its object-centric nature, RSRD can apply learned demonstrations to arbitrary reorientations of the object while transfering demonstration morphology.

**Acknowledgement** This project was funded in part by NSF:CNS-2235013, IARPA DOI/IBC No. 140D0423C0035, and DARPA No. HR001123C0021; Justin Kerr, Chung Min Kim, and Brent Yi are supported by the NSF Research Fellowship Program, Grant DGE 2146752.

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

# A  Appendix

## A.1   Implementation Details

**Part-Centric Feature Fields**   We train a dense feature field over this object to facilitate tracking in the later stage supervising the feature field using each view's DINO feature map as in prior work [67, 66, 79, 69]. Each Gaussian is embedded with a feature vector of dimension $D$, which can be projected onto the DINO space with a small MLP applied per-pixel. The part-centric feature field is trained with a per-pixel MSE loss for 6000 steps, around 3 minutes. The result is an object model which can be differentiably rendered at high framerate (30fps HD), separated into parts with dense feature descriptors whose poses can be differentiated through with respect to pixels and time. We build on Nerfstudio's [80] Splatfacto variant of 3DGS, which includes improvements like camera pose optimization and feature rasterization, using the gsplat [81] rasterization backend.

Our implementation is built on Nerfstudio's Splatfacto model, taking advantage of the same splitting and culling logic. We represent DINOv2 ViT-B/14 features by taking the PCA across all input image features to compress them to 64 dimensions, then assign every gaussian a learnable 32-dimension vector. These vectors can be rasterized with the exact same rendering equations as RGB, using the N-D rasterization implementation from the gsplat library. After rasterization, pixel values are passed through a 4-layer, 64-wide MLP to output the final 64-dimension features. The outputs are supervised with a simple MSE loss against the image features. We additionally apply a nearest-neighbors total-variation loss, which at each step minimizes the standard deviation of a gaussian with its 3 neighbors, encouraging feature embeddings to be spatially smooth. To refine camera poses from their potentially noisy initialization from Polycam, we enable camera optimization from view matrix gradients propagated from RGB rasterization.

**3D Motion Recovery**   To reduce high-frequency noise in the rendered DINO features, we blur them with a kernel equal to the ViT patch size, and clip DINO features which correspond to low alpha in the rendered view, which can correspond to stray floating gaussians. Pose optimization is first done per-frame with no temporal smoothing, iterating 50 steps with the Adam [82] optimizer. We optimize pose offsets represented as quaternions and translations, such that transforms apply relative to each object part centroid. Over the course of the optimization for each frame, the learning rate is initialized high, then decayed by a factor of 5x to allow poses to settle. To improve the temporal coherence of motion recovery, after per-frame tracking we jointly optimize all frames at once with a Laplacian temporal smoothness loss on neighboring poses. See Fig 3 for an illustration.

We use the robust loss $\rho$ proposed in Barron [83], setting $\alpha = 1.0$ for objects whose parts stay attached, and $\alpha = 0.1$ for objects containing separable parts (nerf gun, USB cable), which decreases the strength of the loss for gaussians which deviate far away.

During loss calculation we weight the three optimization objectives with $\lambda_{ARAP} = 0.2$, $\lambda_{MONO} = 0.5$, $\lambda_{DINO} = 1$ before summing. Adam's learning rate is decreased from 0.005 to 0.0005 over the course of 50 steps each frame with an exponential decay. During tracking we sample 30,000 random pairs within the object mask to use with the sparse depth loss, where the object mask is defined by pixels with rendered alpha values over 0.9 (mostly opaque).

**Speed**   We train the part-centric feature field for 6000 steps, which takes about 3 minutes on an RTX 4090 GPU. Prior to this, we train a GARField model to 10000 steps, which takes around 5-10 minutes depending on the number of input images. During tracking, our trajectories consist of 3-4 second video clips at 30fps, and tracking runs at approximately 1.4sec/frame, taking 2-3 minutes total per trajectory. Notably, hand detection account for a substantial portion of the time spent per frame, and without hand tracking can run at approximately 1.2fps. Detecting the object pose in the robot's workspace takes 30 seconds to run the multi-seed optimization search. Computing grasps and motion planning collision-free trajectories takes around 1 minute for single hand demonstrations and 3 minutes for bimanual demonstrations. Speeding up and streamlining the pipeline is a clear opportunity for future work to study.

**Grasp and Motion Planning**  We use an analytic grasp generation method to guarantee grasps on every part, because off-the-shelf grasp planners tend to focus on holistic object grasping rather than part-level. It first converts each group of gaussian centers to a mesh by taking its alpha shape, then smooths and decimates it to produce smooth normals. We sample 20 antipodal grasps axes per part using the grasp procedure described in Mahler et al. [84], then augment them with rotation and and grasp axis translation into 480 grasps which are stored in the part frame for later usage.

As described in main text's section 4.3, part contact selection outputs a ranked list of candidate object parts to interact with, from human hand detection. Then, the planner attempts to find the first set of parts where the motion is executable. For bimanual tasks the list is composed of length-two tuples $[(p_1, p_2), ...]$, one part for each hand, and we exhaustively check over both arms i.e., left arm to $p_1$ and right to $p_2$, and vice versa. We first optimize for the robot motion following the pose of the desired object using a trajectory optimizer implemented via sparse Levenberg-Marquardt in JAX [85], optimizing for smooth joint positions given a set of 6D robot gripper poses, as cuRobo does not provide a waypoint-based trajectory optimization. Then, for the successful trajectories, we use cuRobo to plan collision-free trajectories to the pre-grasp and grasp pose for each part.

## A.2  Experiment Details

**Robot Trials**  Please see the supplemental video for example executions of these motions on the robot, as well as failure case videos.

The experiment motions for each objects are as follows:

1. Red Box: Closing the box, by lowering the lid
2. Nerf Gun: Sliding back the firing mechanism of the gun
3. Scissors: Closing, then opening the scissors
4. Sunglasses: Folding back the left leg of the sunglasses
5. Bear: Waving the right arm of the bear
6. LED Light: un-folding the LED light panel 90 degrees up
7. Wirecutter: closing and opening
8. Stapler: folding the stapler closed from an open position
9. USB Plug: unplugging the usb cable from a power brick

**Tracking Evaluation**  3D pose for part trajectories is manually annotated for keyframes by visualizing the dense RGB-pointcloud obtained from the depth camera in a 3D viewer, then manually moving the rendered gaussian splat of the object part to align with this pointcloud.

