# OpenReview forum: "Robot See Robot Do: Imitating Articulated Object Manipulation with Monocular 4D Reconstruction"
_robot-learning.org/CoRL/2024/Conference — CoRL 2024_

### Official Review · Reviewer_P34h · 2024-07-18
**Nice example of showing how a combination of new tools can support a challenging class of third-person imitation learning problems**

**Originality:** 3
**Technical Quality:** 4
**Clarity Of Presentation:** 3
**Potential Impact:** 4
**Recommendation:** 4
**Confidence:** 4

**Review:**

Strengths:
- High technical quality. Clean idea and approach combining 3D reconstruction, part segmentation, and part tracking.
    - Especially compelling demonstration of not requiring any task-specific training (instead just using existing off-the-shelf tools).
    - Nice insights gained from ablation experiments (e.g., showing the importance of as-rigid-as-possible regularization for part tracking).
- Clarity is quite good. The writing is mostly clear and straightforward to follow. The figures are well done and helpful in aiding understanding. The experiments and results are also presented well, with evidence supporting the main messages and hypotheses.
- Originality is relatively good. It is not quite a "novel method" as it combines existing techniques, but it is still a pretty novel combination of quite new tools from a variety of adjacent literature. Also a relatively new way to cast imitation learning for manipulation from third-person video.
- Potential significance and impact are quite high. Third-person imitation learning for manipulation is a major goal, and this demonstrates one meaningful step toward making it much easier to produce robot motions that realize object motions shown in videos that are widely available on the internet. Especially considering that the paper addresses the complications of operating from single-view videos, RGB observations alone, and objects with individually articulated parts.

Weaknesses:
- (Minor, but a weakness nonetheless) The overall success rates in the real world could be higher. This demonstrates that the method may still have a way to go for being directly useable for applications requiring more reliability.
- As mentioned in the assumptions, the object scan must be taken in the same starting configuration as the demonstration. This seems to mean the current pipeline cannot be directly applied to more in-the-wild videos of objects being manipulated.
- Similarly, the requirement of visible hand poses in the video and using these hand poses to obtain contact points/grasps is somewhat limiting. Not requiring visible hand poses and integrating with other off-the-shelf grasp predictors would perhaps make the approach even more applicable for others to utilize.
- Relatedly, another weakness is the clarity of the section explaining how tracked hand poses are converted to a biased search for grasp poses. I found this part a bit hard to follow.
- (Also minor) There are quite a few moving parts and integration of external tools (i.e., 3DGS, GARField, MonoDepth model, DINO, HaMeR). This is clearly an integral part of the approach, but may potentially make it challenging for users less familiar with each of these tools to get up and running. A well-engineered code release would significantly mitigate this.

**Quality Of The Limitations Section:**

3

**Questions For Rebuttal:**

- Is there a specific reason we cannot run essentially the same pipeline but instead use a more generic grasp detector? Everything else feels pretty “off-the-shelf”, IMO it would be appealing if the grasp planning part was as well.
    - In addition to clarifying why it's used/needed, I would appreciate revising that section in the paper to make it clearer.
    - This is not critical, but it would be nice to add some qualitative evidence showing that the hand pose estimator is not strictly necessary.
- “For initializing the pose during robot deployment, depth from a stereo camera is used instead of monocular depth.” I believe this is the first time mono-depth was mentioned, and felt it was a bit unclear. It is made a bit clearer in later sections, but I would suggest that a clearer mention of where and why monocular vs. stereo depth is used should be made earlier on in the paper.

**Robotics Focus:**

4

**Summary Of Paper:**

This paper aims to learn articulated object manipulation from third-person video demonstrations. The main idea of the approach is to model 3D geometries, articulations, and part trajectories that occur in the demonstration, and then plan robot motions that reproduce the part trajectories. The key insight is to simplify capturing the geometric model and part trajectories with commodity monocular cameras using components like Gaussian splatting for 3D object modeling, GARField for part segmentation, and DINO features for object tracking. The main contributions are a pipeline enabling the challenging task of part trajectory tracking from monocular video and applying the method for bimanual manipulation tasks on real-world objects with no task-specific data or fine-tuning. Experiments show that the method achieves sub-1cm part pose tracking error and that specific design choices such as as-rigid-as-possible regularization and using DINO features are key for effective performance.

**Summary Of Recommendation:**

The paper shows a compelling demonstration of several new tools from robotics, graphics, and computer vision coming together to enable a challenging third-person imitation learning task. The writing, figures, and experimental results clearly articulate and support the primary claims.

---

### Official Review · Reviewer_rPHN · 2024-07-20
**Articulated Object Imitation Learning Method Based on Part-level Gaussian Splatting, Requires Clarification on Method Details**

**Originality:** 3
**Technical Quality:** 4
**Clarity Of Presentation:** 4
**Potential Impact:** 2
**Recommendation:** 3
**Confidence:** 3

**Review:**

The paper has interesting ideas, but it is heavy on engineering. It uses Gaussian splatting for part modeling, Garfield [1] for part decomposition, and HaMeR [2] for hand tracking. The model mainly shines in Object+Part registration and tracking but not as much in the grasping part. This is surprising, as during part decomposition, the method relies on Garfield (which only uses 3D scale information) without using motion heuristics, which most prior work uses [3,4]. Garfield also requires two inputs: scale and 3D point. It can also perform this automatically by selecting a granularity level hierarchically, but in Figure 4, we can see different levels of granularity in part decomposition. Is the part selection/decomposition process manual?

Some additional clarity comments:
* The “Analysis by Synthesis” keyword is unclear. The authors could explain it either in the manuscript or in supplementary material.
* As I understand it, the object is only rotated on the Z-axis and is limited to +-45 degrees. Would this method work in higher degrees? I understand that the object pose initialization process relies heavily on object rotation being close to the original object rotation with respect to the camera pose.
* The initialization of the object pose in the first frame is unclear. The human demonstration video's dino features are based on the camera's pose. You must render the 2D feature field from a virtual camera pose using Gaussian splats to compare to the Dino features. The rendered view should have dino features that have corresponces with the dino features of the human demonstration video. Is this why you are limiting rotation perturbation to 45 degrees?
* In the same paragraph, it would be beneficial to write the process described in lines 163-166 in more detail (ideally with some equations) as it is very ambiguous.

[1] Kim, Chung Min, et al. "Garfield: Group anything with radiance fields." Proceedings of the IEEE/CVF Conference on Computer Vision and Pattern Recognition. 2024.

[2] Pavlakos, Georgios, et al. "Reconstructing hands in 3d with transformers." Proceedings of the IEEE/CVF Conference on Computer Vision and Pattern Recognition. 2024.

[3] Martin, Roberto Martin, and Oliver Brock. "Online interactive perception of articulated objects with multi-level recursive estimation based on task-specific priors." 2014 IEEE/RSJ International Conference on Intelligent Robots and Systems. IEEE, 2014.

[4] Nie, Neil, et al. "Structure from Action: Learning Interactions for 3D Articulated Object Structure Discovery." 2023 IEEE/RSJ International Conference on Intelligent Robots and Systems (IROS). IEEE, 2023.

**Quality Of The Limitations Section:**

3

**Questions For Rebuttal:**

I would ask the authors to clarify my questions regarding how they automatically generate well-selected object parts from Garfield. Secondly, I would ask them to clarify my questions regarding object pose estimation and rotation limitations.

**Robotics Focus:**

4

**Summary Of Paper:**

This paper proposes a method that performs visual imitation learning by tracking part motions through feature fields from a reference human demonstration. Feature Fields are learned through Gaussian splatting, and parts are decomposed from the original object using Garfield [1]. These parts are tracked using the feature fields, and the grasp pose is estimated based on the human hand thumb and index finger pose. Robot performs a similar motition to imitated human demonstration video by following the desired part trajectory.

**Summary Of Recommendation:**

The method has some limitations. But overall, it is a well-demonstrated method.

---

### Official Review · Reviewer_btbD · 2024-07-20
**Very interesting problem statement and method, but needs more study and exeperimentation**

**Originality:** 3
**Technical Quality:** 2
**Clarity Of Presentation:** 3
**Potential Impact:** 3
**Recommendation:** 3
**Confidence:** 3

**Review:**

### Strengths:
1. Very interesting problem statement and method proposed for imitating bimanual human object manipulation from a single demonstration via object part tracking using neural feature fields.
2. Experiments performed give insight into several steps of the method, however, not all, see below.
3. Supplementary video is great and shows both success and failure cases, which I appreciate a lot!
4. Good discussion of limitations of the work as well.

### Weaknesses:
Major weakness is lack of sufficient experimentation, it’s only tested on 5 objects for 5 runs, which is insufficient for a full paper. The results should include more robust testing of the method to prove it’s utility such as testing on novel objects, novel orientations, ablating and justifying each part of the method from the choice of Gaussian splatting, loss terms used, systems design choices, etc. The current state of experiments is demonstration level and which is indeed promising to keep working on this project!

**Quality Of The Limitations Section:**

3

**Questions For Rebuttal:**

1. It is unclear how exactly Gaussian spatting is leveraged? It should be explained in the methods section clearly with equations, followed by how t is it mapped to DINO features. Justification of using Gaussian spatting is also missing in the experiments.
2. How much time does it take for the system to learn a new task/demonstration, ie, doing all the steps of the learning process? Is there any manual annotation needed for any of the steps? Which steps and losses are optimized prior to execution, and which ones are done during execution? - these questions need to be made clear in the methods section, currently it is not very obvious.
3. Can this system learn and solve many tasks instead of learning one at a time, ie, once demonstrated can maintain the learning for future execution, because if not, then this kind of system might not be useful if the human always needs to keep demonstrating. This needs to be verified experimentally.

**Robotics Focus:**

4

**Summary Of Paper:**

Imitating bimanual human object manipulation from a single demonstration via object part tracking using neural feature fields.

**Summary Of Recommendation:**

I think the paper addresses an interesting problem, however needs more study and experiments, which is why I think the authors should resubmit later with complete results.

---

### Author Rebuttal · Authors · 2024-08-09

Thanks all reviewers for their constructive feedback, please see the revised draft here!

---

### Decision · Program_Chairs · 2024-09-04

**Decision:**

Accept

**Comment:**

The reviewers found strengths in this submission, but also clearly articulated some questions for the rebuttal phase.
Their initial assessments were quite divergent, so the response phase helped.